# Image-based parameter inference for epithelial mechanics

**Goshi Ogita**[ID][1,2], **Takefumi Kondo**[ID][2], **Keisuke Ikawa**[ID][1,3], **Tadashi Uemura**[ID][2], **Shuji Ishihara**[ID][4,5]*, **Kaoru Sugimura**[ID][1,3,5,6]*

**1** Department of Biological Sciences, Graduate School of Science, The University of Tokyo, Tokyo, Japan, **2** Graduate School of Biostudies, Kyoto University, Kyoto, Japan, **3** Institute for Integrated Cell-Material Sciences (WPI-iCeMS), Kyoto University, Kyoto, Japan, **4** Graduate School of Arts and Sciences, The University of Tokyo, Tokyo, Japan, **5** Universal Biology Institute, The University of Tokyo, Tokyo, Japan, **6** Department of Computational Biology and Medical Sciences, Graduate School of Frontier Sciences, University of Tokyo, Chiba, Japan

* csishihara@g.ecc.u-tokyo.ac.jp (SI); sugimura@bs.s.u-tokyo.ac.jp (KS)

## Abstract

Measuring mechanical parameters in tissues, such as the elastic modulus of cell-cell junctions, is essential to decipher the mechanical control of morphogenesis. However, their *in vivo* measurement is technically challenging. Here, we formulated an image-based statistical approach to estimate the mechanical parameters of epithelial cells. Candidate mechanical models are constructed based on force-cell shape correlations obtained from image data. Substitution of the model functions into force-balance equations at the cell vertex leads to an equation with respect to the parameters of the model, by which one can estimate the parameter values using a least-squares method. A test using synthetic data confirmed the accuracy of parameter estimation and model selection. By applying this method to *Drosophila* epithelial tissues, we found that the magnitude and orientation of feedback between the junction tension and shrinkage, which are determined by the spring constant of the junction, were correlated with the elevation of tension and myosin-II on shrinking junctions during cell rearrangement. Further, this method clarified how alterations in tissue polarity and stretching affect the anisotropy in tension parameters. Thus, our method provides a novel approach to uncovering the mechanisms governing epithelial morphogenesis.

## Author summary

Thanks to recent advances in imaging techniques, researchers can track various events that occur during development, such as collective cell movement, morphogen signaling, and transcription from specific genome loci. A new challenge is to extract biologically important information from large and complex image data. Here, we developed an efficient and rapid method to extract mechanical parameters from image data, in which epithelial cell shape is visualized. A test using artificially generated data showed that the proposed method provided accurate estimates of parameters without performing numerical simulations. This is in contrast with previous methods that used summary statistics. By applying this method to *Drosophila* epithelial tissues, we uncovered that junction

**Data Availability Statement:** The authors declare that the data supporting the findings of this study are available within the paper and its Supplementary files. The code for parameter inference and model selection can be downloaded

from https://github.com/Sugimuralab/Image BasedParameterInferenceForEpithelialMechanics. The code for Bayesian force inference can be downloaded from https://github.com/IshiharaLab/BayesianForceInference.

**Funding:** This study was financially supported by JSPS KAKENHI Grant (17K15125) to K.S., JSPS KAKENHI Grant (18H01185) to S.I. and K.S., JSPS JRPs with SNSF (JPJSJPR 20191501) to S.I. and K.S., AMED PRIME (20gm5810025h9904) to K.S., and JST CREST (JPMJCR1923) to S.I. K.I.'s salary was supported by AMED and JST. The funders had no role in study design, data collection and analysis, decision to publish, or preparation of the manuscript.

**Competing interests:** The authors have declared that no competing interests exist.

remodeling and resistance to extrinsic pulling are controlled via different tension parameters. Moreover, this method could detect changes in anisotropic tension parameters in genetically or surgically manipulated tissues. We anticipate that the proposed method will provide a powerful tool to dissect the mechanical control of morphogenesis.

## Introduction

Mechanics play many roles during morphogenesis [1–5]. The mechanical properties of cells and tissue determine whether and how tissue flows upon active force generation by cells or pushing/pulling from neighboring tissues [6,7]. Cells resist applied forces such that morphogenesis proceeds without disrupting the structural integrity of tissue [8]. Moreover, tissue stress feeds back on protein distribution and activity to tune gene expression and biochemical signaling [9,10]. Collectively, tissue shape emerges from the complex interplay between mechanics and genetics.

Mechanical measurements are therefore key to deepening our understanding of morphogenesis. *In vivo* force/stress measurement techniques have advanced greatly in the last decade [11–14]. For instance, in a monolayer epithelium, which is one of the most studied experimental systems in tissue mechanics, laser ablation, optical tweezers, and force inference have clarified that cells increase tension on specific junctions to trigger cell rearrangement (Figs 1A and 1B and 2) [15–19]. Other methods, including atomic force microscopy, FRET tension sensors, and liquid droplets, have been successfully applied to probe forces and stresses in various developmental contexts. In contrast, *in vivo* measurements of mechanical properties (*e.g.*, the elastic modulus of cells) are still technically challenging. Classical methods such as micropipette aspiration and parallel-plate compression are limited to tissues where direct contact with a sample is possible [20,21]. More recently developed methods are contact-free but require sophisticated expertise [22–24].

To compensate for the difficulty of *in vivo* mechanical measurement, modeling is widely used to analyze the relationship between cell mechanics and morphogenesis [25–27]. For instance, the cell vertex model (CVM) computes the position and connectivity of cell vertices via the minimization of virtual work, by which the balance of junction tensions and cell pressure is considered (Fig 1C, Materials and Methods) [28–30]. Many studies have been conducted to investigate how changes in the magnitude and direction of cellular forces and the mechanical properties of cells, which are implemented by controlling mechanical parameters, result in different tissue shapes (reviewed in [29,30]). The biological relevance of these modeling results obviously depends on the choice of model equations and parameters. Therefore, an efficient and rapid method for the data-driven determination of model equations and parameters is needed.

Previous studies used summary statistics such as the polygonal distribution of cells, the average area in each polygonal class, and recoil velocity upon laser ablation of cell junctions to search for parameter values that recapitulate *in vivo* tissue. Specifically, numerical simulations were repeated with different parameter sets, and the data obtained by the simulation were compared with experimental data via summary statistics. The parameter set that best fit the experimental data was selected by manual fitting, a least-squares method, or approximate Bayesian computation [31–35]. These approaches were indirect in the sense that they did not directly consider model equations for parameter fitting. Moreover, different forms of model equations have not been tested based on statistical criteria.

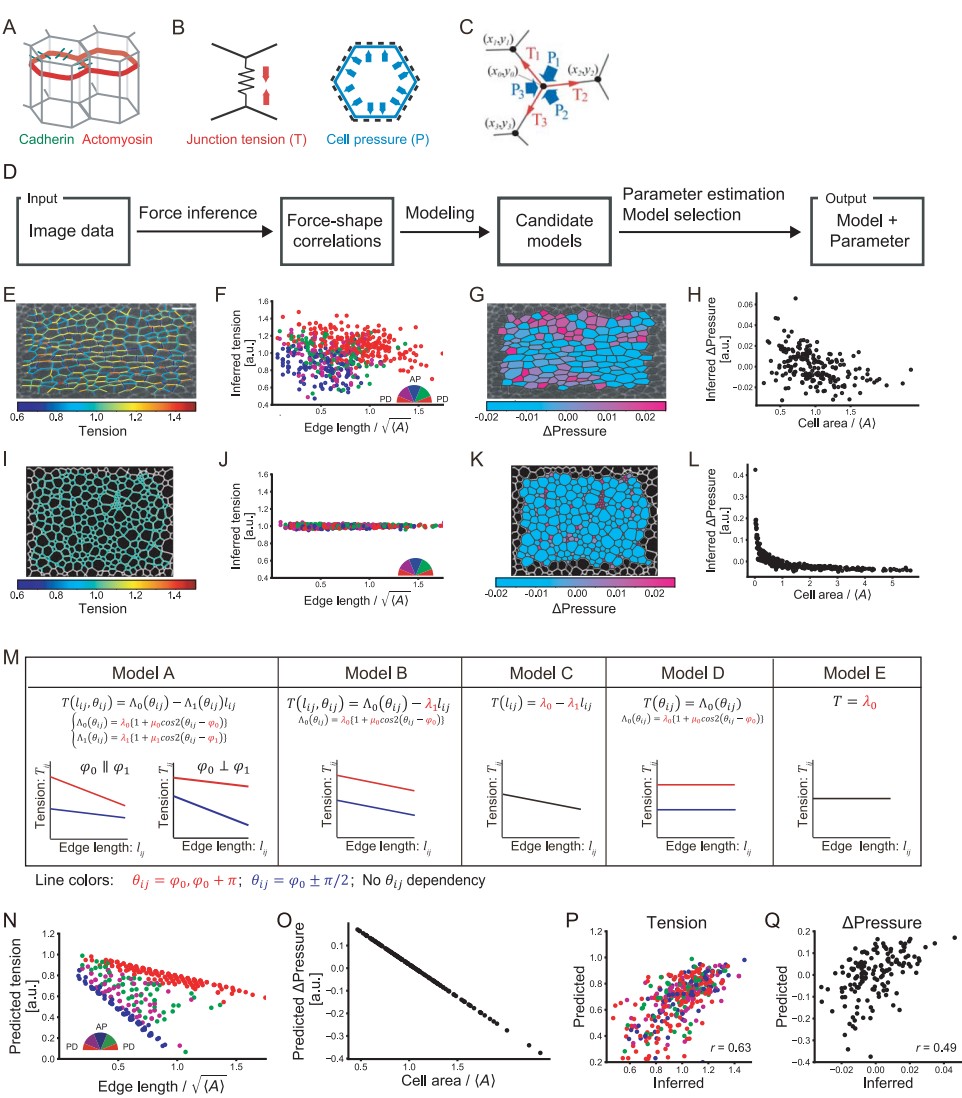

**Fig 1. Image-based parameter estimation for epithelial mechanics.** (A) The structure of the monolayer epithelium. Epithelial cells adhere to each other via cell adhesion molecules such as cadherin (green). Actomyosin cables (red) run along the cell cortex in the plane of the adherens junction (AJ). (B) Forces acting along the AJ plane. Left: Junction tension. Right: Cell pressure. (C) Balance of tensions (T) and pressures (P) at a cell vertex indicated by a black dot. (D) Flowchart of the proposed method to estimate mechanical parameters from image data, in which cell contour is visualized. (E–L) Force-shape correlations obtained by Bayesian force/stress inference in epithelial tissue (E–H) and in foam (I–L). Bayesian force/stress inference yields maps of junction tension (E, I) and cell pressure (G, K), which are overlaid on input images (epithelial tissue: DE-cad-GFP in *Drosophila* pupal wing at 22 h APF, foam: a coarsening foam [73]). Each junction is classified by its orientation relative to the horizontal axis (semicircle; PD: proximal-distal and AP: anterior-posterior), and its inferred tension is plotted against its length (F, J). The inferred pressure is plotted against the cell/foam area (H, L). $\langle A \rangle$ represents the average cell area. (M) Table of a series of models, which are designated model A to model E in order of decreasing complexity. A schematic showing the dependence of tension on junction length and orientation and the tension function is shown for each model. Parameters are indicated in red. (N–Q) Results of parameter estimation from the *Drosophila* wing image shown in E. (N) The orientation of each junction relative to the PD axis of the wing is classified (semicircle), and its tension predicted from estimated parameters is plotted against its length. (O) Cell pressure predicted from estimated parameters is plotted against cell area. (P, Q) Correlation between predicted force values obtained by the parameter estimation method and inferred force values obtained by Bayesian force/stress inference. Predicted tensions and pressures (vertical axis) are plotted against inferred tensions and pressures (horizontal axis), respectively. The correlation coefficient is shown in the bottom right corner. Scale bar: 10 μm in (E).

In this study, we developed a data-driven parameter estimation method for epithelial mechanics (Fig 1). Starting from image data in which cell contours are visualized, we took advantage of Bayesian force/stress inference [36] to perform batch quantification of force-cell shape correlations. Based on this information, we constructed candidate model equations (tension function $T(l,\theta)$ and pressure function $P(A)$ explained below). Substitution of the model functions into the force-balance equations leads to an equation with respect to the parameters of the model, by which one can estimate the parameter values using a least-squares method. We confirmed the accuracy of parameter estimation and model selection in artificially generated data. Our method identified tissue- and stage-specific changes in parameter values in *Drosophila* epithelial tissues. Based on these data, we uncovered that junction remodeling and resistance to extrinsic pulling are controlled via different tension parameters. Furthermore, we showed that this method could detect how genetic or surgical modification affects mechanical parameters. The simplicity of the proposed method would allow an efficient and rapid analysis of mechanical parameters *in vivo*, thereby helping us quantitatively dissect the mechanisms of morphogenesis.

## A mechanical parameter estimation method

In this section, we explain the formulation of the mechanical parameter estimation method (Fig 1D). First, we will show a representative example of the correlation between junction tensions/cell pressures and cell shape features that is obtained by Bayesian force/stress inference (Fig 1E–1L; Materials and Methods) [36]. Next, we list candidate tension functions $T(l,\theta)$ and a pressure function $P(A)$ based on the observed force-cell shape correlations (Fig 1M). Finally, we explain the formulation of the proposed method, by which mechanical parameters in tension and pressure functions are estimated by solving overdetermined force-balance equations at the cell vertices.

### Quantification of force-shape correlations from image data

Associating cellular forces with cell shape and geometry provides information about material mechanical properties and a mechanical model of the tissue. However, the number of data points required to quantitatively assess force-shape correlations in a tissue is expected to be at least several dozens for each sample. Obtaining such a large amount of experimental data requires significant time and effort. Bayesian force/stress inference, which yields maps of junction tension and cell pressure, enables the quantitative assessment of force-shape correlations in a tissue.

We applied Bayesian force/stress inference to image data of *Drosophila* epithelial tissues and examined the correlation between junction tensions and different kinds of cell shape features. Our analysis revealed that inferred tensions are correlated with the junction orientation and length in the *Drosophila* pupal wing at 22 hours after puparium formation (h APF) (Fig 1E and 1F; See Fig 2A for the schematic of pupal wing). Tensions are larger for red proximal-distal (PD) junctions than blue anterior-posterior (AP) junctions (compare red and blue dots in Fig 1F). Within the same angular class of junctions, tensions are negatively correlated with junction length. Moreover, tensions on blue AP junctions exhibit a stronger dependence on junction length. In sharp contrast, the same plot obtained from a foam image shows that inferred tension is constant with respect to the interface length and direction, consistent with constant tension in foam (Fig 1I and 1J). These results suggest that material mechanical properties can be characterized by force-shape correlations and that cell junctions behave as anisotropic springs.

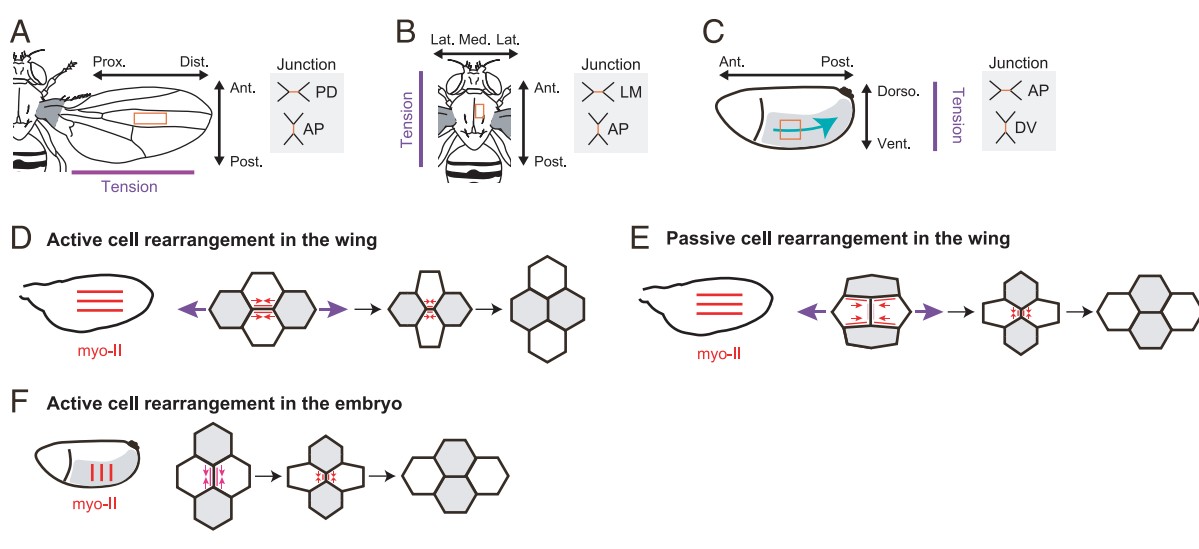

**Fig 2. *Drosophila* epithelial tissues examined in this study.** (A–C) Schematics of the adult fly (adapted from [69]) (A, B) and of the fly embryo at the germband elongation (GBE) stage (C). Orange rectangles indicate the regions studied in this study. Purple lines indicate the orientation of tissue tension. Schematics of junction are shown in the right. (A) The hinge is shaded gray. In this and all subsequent figures showing the wing, the vertical and horizontal directions are aligned with the AP and PD axes, respectively. (B) In this and all subsequent figures showing the notum, the vertical and horizontal directions are aligned with the AP and ML axes, respectively. (C) In this and all subsequent figures showing the embryo, the vertical and horizontal directions are aligned with the DV and AP axes, respectively. A cyan arrow indicates cell flow during GBE. (D–F) Schematics showing active cell rearrangement in the pupal wing (D), passive cell rearrangement in the pupal wing (E), and active cell rearrangement in the embryo at the GBE stage (F). In the left panels, red lines indicate the global myo-II polarity in tissues. In the right panels, red/pink arrows and purple arrows indicate junction tension and extrinsic pulling forces from the hinge, respectively. Line color represents the local myo-II concentration along the junction. Cells localize myo-II on PD junctions to resist extrinsic pulling forces from the hinge (red lines in D). The myo-II enrichment along the remodeling junctions is indicated by the change in line colors from pale pink to red along the vertical shrinking junction in E and from dark pink to red along the vertical shrinking junction in F.

## Model construction

Based on the observed relationship between inferred tension and cell shape features, we constructed a candidate tension function as

$$T(l_{ij}, \theta_{ij}) = \Lambda_0(\theta_{ij}) - \Lambda_1(\theta_{ij})l_{ij}, \tag{1}$$

where $l_{ij}$ and $\theta_{ij}$ are the length and orientation of the cell junction between the $i$-th and $j$-th cells, respectively. $\Lambda_0(\theta_{ij})$ and $\Lambda_1(\theta_{ij})$ are given by

$$\begin{cases} \Lambda_0(\theta_{ij}) = \lambda_0\{1 + \mu_0 cos2(\theta_{ij} - \varphi_0)\} \\ \Lambda_1(\theta_{ij}) = \lambda_1\{1 + \mu_1 cos2(\theta_{ij} - \varphi_1)\}. \end{cases} \tag{2}$$

$\Lambda_0(\theta_{ij})$ represents the anisotropic line tension. The magnitude and orientation of its anisotropy are determined by $\mu_0$ and $\varphi_0$. $\Lambda_1(\theta_{ij})$ represents the anisotropic spring constant of the junction, which can be regarded as feedback between myosin-II (myo-II)-generated junction tension and junction shrinkage. A sign of $\Lambda_1(\theta_{ij})$ determines whether the feedback is positive (*i.e.*, accelerating junction shortening/lengthening) or negative (*i.e.*, stabilizing the junction length). $\mu_1$ and $\varphi_1$ represent the magnitude and orientation of anisotropy in the feedback, respectively. Such feedback between tension/myo-II and junction shrinkage is suggested by experimental observations that myo-II is getting concentrated specifically on shrinking junctions during cell rearrangement (the increase in myo-II level during junction shrinkage was quantified in [37–39]) and has been studied in the context of *Drosophila* embryogenesis [39–41]. In addition to

the model described above, simpler models were considered, as summarized in Fig 1M. Hereafter models are designated as model A to model E in order of decreasing complexity, which is defined by the number of parameters [42,43]. Model A, which employs Eq 2, is a full model and is categorized as an anisotropic spring model. Model B omits the anisotropy in the spring constant of the junction by putting $\mu_1 = 0$. In model C, junction tension is assumed to depend only on the junction length but not on the junction orientation by putting $\mu_0 = \mu_1 = 0$. In model D, only line tension is considered by eliminating $\Lambda_1(\theta_{ij})$. In model E, junction tension is assumed to be constant irrespective of the junction length and orientation (i.e., $T(l_{ij}, \theta_{ij}) = \lambda_0$).

As cell pressure is negatively correlated with cell area (Fig 1G and 1H), we followed previous studies and set $P(A_i) = -kA_i + P_0$, where $A_i$ is the area of $i$-the cell, $k$ is the elastic modulus of cells, and $P_0$ is the basal pressure [15,31]. We tested other forms of pressure functions and found that the results in this study were not significantly affected by the choice of pressure function (S1 Fig; Discussion).

For comparison, we also tested a widely used model of epithelial mechanics (designated the conventional model hereafter) [29,30]. The conventional model sets a tension function so that line tension and cortical elasticity are incorporated as

$$T(L_i, L_j, \theta_{ij}) = \Lambda_0(\theta_{ij}) + \Gamma(\theta_{ij})(L_i + L_j), \tag{3}$$

where $L_i$ and $L_j$ are the cell perimeter of $i$-th and $j$-th cells, respectively. $\Lambda_0(\theta_{ij})$ and $\Gamma(\theta_{ij})$ are defined in the same way as in Eq 2 to represent the anisotropy of line tension and cortical elasticity. We also considered three simpler models, where the anisotropy in the line tension and/or cortical elasticity are omitted. The pressure function is the same as above.

## Parameter estimation by solving force-balance equations

The parameter values of the mechanical models were estimated by solving force-balance equations over the epithelial tissue. We approximated an epithelial tissue as a two-dimensional sheet composed of cells represented by polygons. Suppose that the pressure of the $i$-th cell is $P_i$, and the tension of the cell junction between the $i$-th and $j$-th cells is $T_{ij}$. Under the approximations of the cell geometry, the net forces at the 0-th vertex in Fig 1C, $\boldsymbol{f}_0 = (f_0^x, f_0^y)$, are given by the following equations.

$$f_0^x = a_1^x T_{1X} + a_2^x T_{2X} + a_3^x T_{3X} + b_1^x P_1 + b_2^x P_2 + b_3^x P_3,$$

$$f_0^y = a_1^y T_{1X} + a_2^y T_{2X} + a_3^y T_{3X} + b_1^y P_1 + b_2^y P_2 + b_3^y P_3.$$

Here, the coefficients of tensions are $a_1^x = (x_1 - x_0)/||\boldsymbol{x_1} - \boldsymbol{x_0}|| = \cos(\theta_1)$, $a_1^y = (y_1 - y_0)/||\boldsymbol{x_1} - \boldsymbol{x_0}|| = \sin(\theta_1)$, etc., while those of pressures are $b_1^x = (y_2 - y_1)/2$, $b_1^y = -(x_2 - x_1)/2$, etc. In the same way, the net forces for all vertices in the epithelial tissues, $\boldsymbol{F} = (f_0^x, f_0^y, f_1^x, f_1^y, \ldots)^T$, are represented as follows:

$$\boldsymbol{F} = C\boldsymbol{X},$$

where $C$ is an observable $n \times m$ matrix composed of coefficients $a$ and $b$, and $\boldsymbol{X} = (\boldsymbol{T}, \boldsymbol{P})^T$ is an $m$-dimensional vector composed of $T_{ij}$ and $P_i$ [36]. $n$ is the number of force balance equations and equals twice the number of vertices. $m$ is the sum of the number of junctions and the number of cells. On the assumption that epithelial morphogenesis is a quasi-static process where tensions and pressures are nearly balanced ($\boldsymbol{F} \approx \boldsymbol{0}$), the following equation holds:

$$C\boldsymbol{X} = \boldsymbol{0}. \tag{4}$$

For a given mechanical model considered in this study, the junction tensions $T_{ij}$ and cell pressures $P_i$ are represented by functions of junction length $l_{ij}$, angle $\theta_{ij}$, and cell area $A_i$, as $T_{ij} = T(l_{ij}, \theta_{ij}; \boldsymbol{\beta})$ and $P_i = P(A_i; \boldsymbol{\beta}')$, respectively, where $\boldsymbol{\beta}$ and $\boldsymbol{\beta}'$ are a set of model parameters. Thus, substitution of these functions into Eq 4 leads to an equation with respect to the parameters of the model, by which one can estimate the parameter values using a least-squares method. The details of the implementation are explained in the Materials and Methods.

Note that for the parameter estimation to work properly, the geometry of cells, such as the cell junction length and orientation and the cell area, needs to significantly vary among cells in input image data. An illustrative example is the case of a regular hexagonal pattern, for which the force-balance equations at each vertex become identical, and the input data are thereby effectively reduced to a single point. Under such conditions, it is impossible to estimate model parameters from data.

Fig 1N and 1O show the results of parameter estimation from image data of the *Drosophila* pupal wing. The tensions and pressures calculated from the estimated parameters (predicted tensions and pressures) exhibited reasonable correlations with those inferred by Bayesian force/stress inference (Fig 1P and 1Q).

## Model selection

We introduced different forms of tension functions based on the observed correlation between inferred tension and junction length (Fig 1M). These models differ in their complexity, *i.e.*, the number of fitting parameters. To select among models that are based on different tension functions, we employed the Akaike information criterion (AIC) [42,43]. In the problem considered in this study, AIC is defined as

$$AIC = n \times \log(2\pi\hat{\sigma}^2) + n + 2(p + 1), \tag{5}$$

where $p$ represents the number of model parameters. $\hat{\sigma}^2 = \boldsymbol{F}(\hat{\boldsymbol{\beta}})^T \boldsymbol{F}(\hat{\boldsymbol{\beta}})/n$ is the mean square error of the estimation, where $\hat{\boldsymbol{\beta}}$ is the vector of the estimated parameters of the model. For the image data shown in Fig 1E, AIC took the minimum value when model A was used for estimation. Model A was thus selected among the models tested.

## Data sets

### Synthetic data set

In this study, synthetic data sets were used to confirm the accuracy of parameter estimation and model selection. To obtain synthetic data on cell configuration, numerical simulations of the CVM were performed in which the positions and connectivity of vertices $\{\boldsymbol{x}_k\}$ were changed according to the relaxation of virtual work $U_0(\{\boldsymbol{x}_k\})$ [28–31]. The virtual work of the anisotropic spring model is given as

$$U_0(\{\boldsymbol{x}_k\}) = \sum_{[ij]:junction} \left\{ \Lambda_0(\theta_{ij}) l_{ij} - \frac{1}{2} \Lambda_1(\theta_{ij}) l_{ij}^2 \right\} + \sum_{i:cell} \frac{k}{2} (A_i - A_0)^2,$$

where $A_0$ is the target area of a cell. $\Lambda_0(\theta)$ and $\Lambda_1(\theta)$ are defined as Eq 2. The virtual work of the conventional model is given as

$$U_0(\{\boldsymbol{x}_k\}) = \sum_{[ij]:junction} \Lambda_0 l_{ij} + \sum_{i:cell} \frac{1}{2} \Gamma L_i^2 + \sum_{i:cell} \frac{k}{2} (A_i - A_0)^2,$$

where the cell contour length $L_i$ is the sum of the associated cell junction lengths $l_{ij}$.

Differentiations of $U_0(\{x_k\})$ by $l_{ij}$ and $A_i$ lead to tension $T_{ij} = \partial U_0/\partial l_{ij} = T(l_{ij}, \theta_{ij})$ and cell pressure $P_i = -\partial U_0/\partial A_i = P(A_i)$, respectively.

In the simulation, we solved equations

$$\frac{d\boldsymbol{x}_k}{dt} = -\frac{\partial U_0(\{\boldsymbol{x}_k\})}{\partial \boldsymbol{x}_k} = -T_{ij}\frac{\partial l_{ij}}{\partial \boldsymbol{x}_k} + P_i\frac{\partial A_i}{\partial \boldsymbol{x}_k},$$

where derivatives of anisotropy coefficients such as $\partial \Lambda_0(\theta_{ij})/\partial \boldsymbol{x}_k$ are ignored [44]. The numerical simulations were implemented by C++. An initial cell configuration was a 20×20 cell tile, which was generated from randomly distributed centroids with a mean cell area of $A_0$. We adopted a free boundary condition. The discretized time step was set to $\Delta t = 0.1$, and the simulation was run until a configuration of cells became stable ($t = 5000$). To implement a T1 transition, junctions shorter than $0.05 \times \sqrt{A_0}$ were allowed to perform a T1 transition if this transition reduced the virtual work [31]. Data were generated by using a reference parameter set $(\lambda_0, \mu_0, \lambda_1, \mu_1, k) = (0.2, 0.15, 0.03, 0.5, 1.0)$. Different data sets were also generated by altering one of the parameters in the reference parameter set as follows: $\lambda_0 = 0.1, 0.15, 0.25, 0.3$; $\mu_0 = 0.05, 0.1, 0.2, 0.25, 0.3$; $\lambda_1 = 0.01, 0.02, 0.04$; $\mu_1 = 0.1, 0.2, 0.3, 0.4$; $k = 0.75, 2.0, 4.0$. The target cell area $A_0$ was fixed as $A_0 = 1.0$.

### *In vivo* data set

In this study, we analyzed how the mechanical parameters of cells change over the time course of the development of the *Drosophila* pupal wing, notum, and embryo. Here, we summarize what is known about the morphogenesis and mechanics of the three experimental systems.

## Pupal wing

The pupal wing is stretched along the PD axis by forces generated in the hinge (Fig 2A) [45–49]. Wing cells resist the extrinsic pulling force by localizing myo-II on the PD junctions and thereby generating strong junction tension [46]. The magnitude of tension anisotropy peaks at 21–23 h APF [46,48]. It has been shown that cells first undergo active cell rearrangement, by which the PD junction shrinks and is remodeled to form a new anterior-posterior (AP) junction at 15–21 h APF (Fig 2D) [48,50]. At 21–22 h APF and afterward, cells shorten AP junctions, intercalate along the PD axis, and concomitantly adopt a more isotropic cell shape as a means of passive relaxation following axial pulling from the hinge (Fig 2E) [45–48,50].

## Pupal notum

The pupal notum undergoes spatiotemporally heterogeneous yet highly regulated morphogenesis, as indicated by the symmetry relative to the midline axis [16,50]. In a region close to the midline in the scutum (rectangle in Fig 2B), junction tension and myo-II distribution exhibit moderate AP-aligned anisotropy, and cell elongation and cell rearrangement are oriented along the AP axis (Fig 2B) [36,50,51].

## Embryo

Downstream of the AP patterning of the body, which is sequentially specified by transcriptional regulators such as the pair-rule gene *runt*, cells in the germband localize myo-II along the dorsoventral (DV) junctions and induce apical actomyosin flow toward the DV junctions (Fig 2C and 2F) [37,52–54]. As a result, junction tension is elevated at the DV junction, which results in directional cell rearrangement, leading to constriction-elongation of the germband (Fig 2F) [15,18,19,55,56].

## Results

### The proposed method gives accurate estimates of parameters *in silico*

To validate the proposed method, we evaluated the accuracy of the parameter estimation by using synthetic data, where the true values of parameters are available by construction. Numerical simulation was run based on the cell vertex model (CVM), whose virtual work is derived from model A (Materials and Methods). From synthetic data on 2D polygonal cell tiles, the parameters in model A were estimated (Fig 3A). Note that the estimated values are normalized by using the true value of target area $A_0$, since Eq 4 considers only relative values of tensions and pressures. As shown in Fig 3B–3E, the estimation error was very small, with a maximum error of 5.4% in absolute value (the uncertainty of parameter estimation was evaluated in S2 Fig). For synthetic data generated by the conventional model with a typical parameter set ($\Lambda_0$, $\Gamma$) = (0.12, 0.04) [31,35], the estimation errors of $\Lambda_0$ and $\Gamma$ were 0.27% and -0.20%, respectively, indicating that our parameter estimation method works for a different form of model functions.

Next, we assessed the performance of the model selection. Synthetic data were generated by using each of five models that were almost nested except for model D (Fig 1M). For each synthetic data set, parameter estimation was conducted by using the five different models, and the AIC value was calculated. We found that AIC took the minimum value when the data generation model was used for estimation (S3 Fig), indicating that the model selection based on AIC works properly. Taken together, our data confirm that the proposed method can select a proper model and give accurate estimates of mechanical parameters *in silico*.

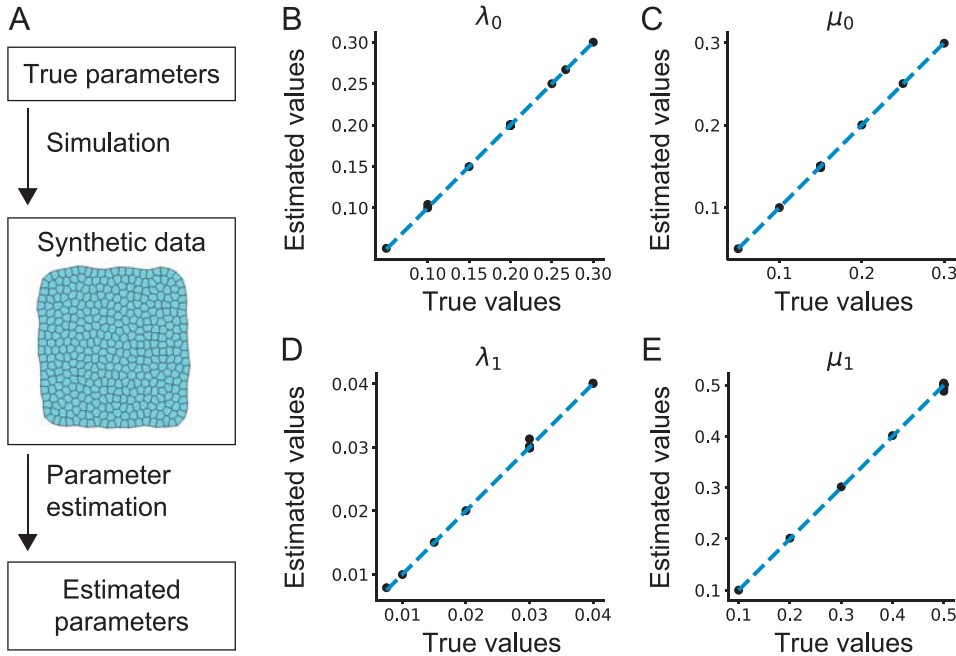

**Fig 3. *In silico* validation of the parameter estimation method.** (A) Schematic of a perfect model experiment. (B–E) Estimated values of parameters are plotted against their true values for $\lambda_0$ (B; the line tension), $\mu_0$ (C; the anisotropy in the line tension), $\lambda_1$ (D; the spring constant of junction), and $\mu_1$ (E; the anisotropy in the spring constant of junction). Estimated values of $\lambda_0$ and $\lambda_1$ were normalized by $k$ and $A_0 = 1.0$ as $\lambda_0 = \hat{\lambda}_0 / \hat{k} A_0^{1.5}$, $\lambda_1 = \hat{\lambda}_1 / \hat{k} A_0$, where $\hat{q}$ denotes the estimated value of $q$. A dashed line indicates $y = x$.

## Parameter estimation shows adequate robustness to image processing error

Both Bayesian force/stress inference and the parameter estimation method use the vertex positions and connectivity obtained from image data as inputs. Thus, it is important to check whether and how the outcome of the methods is affected by error in image segmentation. We previously reported that Bayesian force/stress inference is robust to image processing error [36,51]. Here, we evaluated the noise resistance of the parameter estimation method by adding noise to vertex positions in synthetic data and in segmented images of the *Drosophila* pupal wing, pupal notum, and germband. The deviation of the estimated parameter values in noised data sets was within 25% for most of the cases evaluated (Fig 4). We also confirmed that the deviation of the estimated parameter negligibly affected the predicted values of tension ($r > 0.97$ for all of the noised data; Materials and Methods). These results suggest that our parameter estimation method possesses adequate robustness to image processing error.

## Tension anisotropy in *Drosophila* epithelial tissues is recovered by parameter estimation

To confirm the validity of our parameter estimation method *in vivo*, we examined whether the estimated parameters were consistent with the previously reported tension anisotropy during the development of the *Drosophila* pupal wing, notum, and embryo (see Experimental systems and references therein). We first conducted the parameter estimation and substituted estimated parameter values into the tension function of a selected model (*e.g.*, Eq 1 for model A) to obtain the predicted values of tension. Fig 5A–5L show the dependence of the predicted junction tension on the junction length and orientation. Junction tension appeared to be negatively correlated with the junction length in the pupal wing, pupal notum, and embryo. In addition, the magnitude of tension anisotropy varied among developmental stages and tissues (compare red and blue dots in each plot). To quantitatively characterize the difference, we calculated the tension anisotropy $R_T$ (Materials and Methods) [46]. In the pupal wing and notum, developmental changes in the magnitude and orientation of $R_T$ agreed with those obtained by Bayesian force/stress inference (Fig 5M and 5N and 5P and 5Q; S3B Fig in [46]). In embryos, $R_T$ pointed to the DV axis and became larger at the germband extension stage (GBE), which is consistent with previous reports (Fig 5O and 5R) [15,18,19]. In summary, the proposed method can recover the known tension anisotropy in all tissues examined.

## Positive feedback between junction tension and shrinkage strengthened at specific developmental stages

The pupal wing, pupal notum, and embryo undergo distinct developmental changes in morphogenetic processes and tension patterns, as explained above. Thus, to investigate the relationship between tissue morphogenesis/mechanics and cell mechanical parameters, we closely examined estimated parameter values at different developmental stages and in different tissues (Figs 6 and S4).

The anisotropy in line tension ($\mu_0$ and $\varphi_0$) had no specific orientation at 13–14 h APF in the wing. $\mu_0$ increased and $\varphi_0$ became aligned to the PD axis at 16.5–18.5 h APF (Fig 6A and 6D). $\mu_0$ and $\varphi_0$ remained almost constant afterward until 32 h APF. $\mu_0$ remained low and $\varphi_0$ remained biased toward the AP axis during the development of the notum (Fig 6B and 6E). $\mu_0$ increased along the DV axis in the germband elongation stage in the embryo (Fig 6C and 6F). Collectively, the results suggested that $\mu_0$ and $\varphi_0$ control the magnitude and orientation of tension anisotropy, as expected.

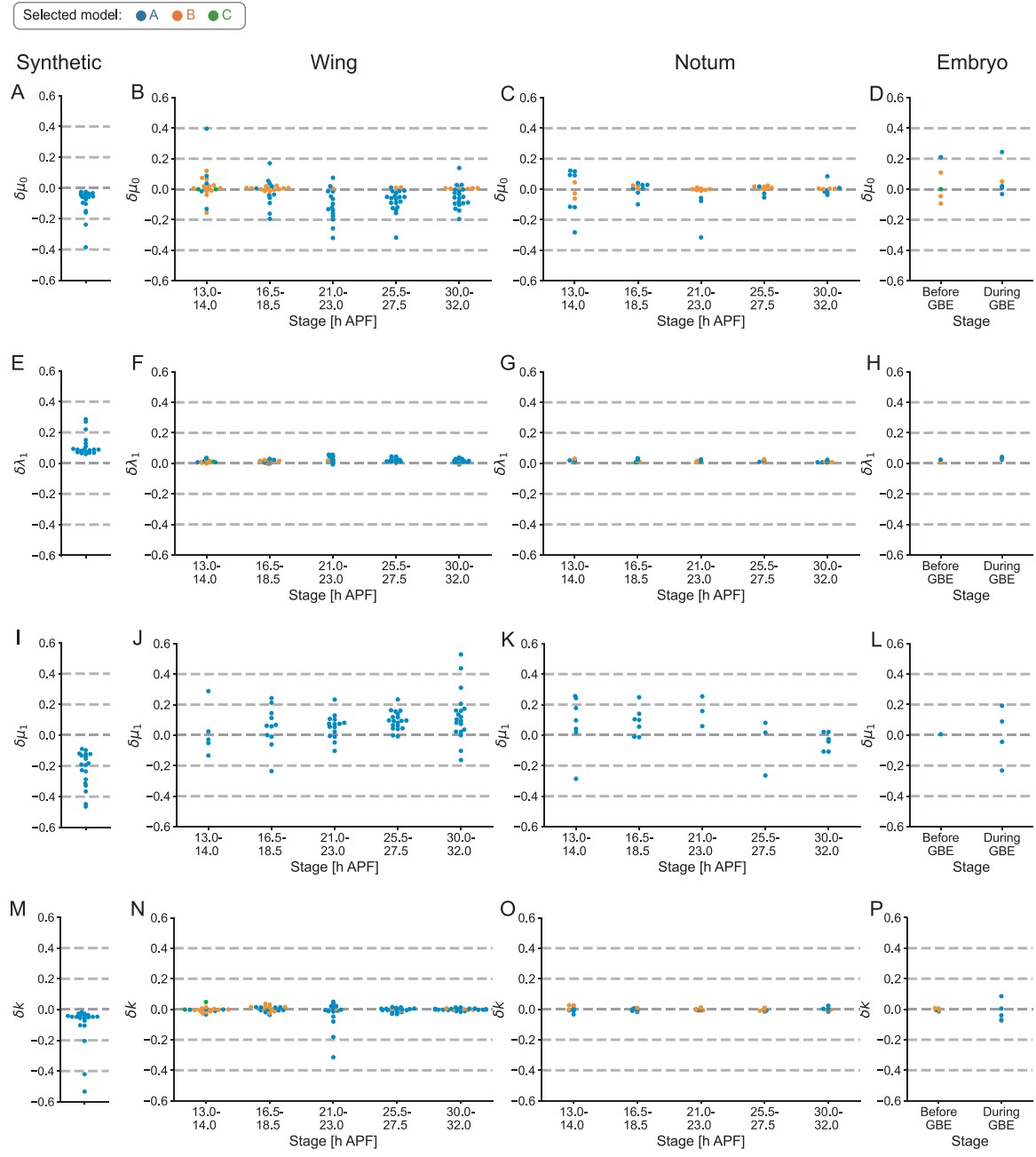

**Fig 4. Evaluation of robustness to image processing error.** (A–P) A noise resistance test on the estimation of $\mu_0$ (A–D; the anisotropy in the line tension), $\lambda_1$ (E–H; the spring constant of junction), $\mu_1$ (I–L; the anisotropy in the spring constant of junction) and $k$ (M–P; the elastic modulus of cells). Parameters were estimated from the original data and noised data sets. Each dot shows the median value of the deviations of estimated parameters in 100 noised data sets (Materials and Methods) for each synthetic data (A, E, I, M) or each input image at the stage indicated in the pupal wing (B, F, J, N), pupal notum (C, G, K, O), and embryo (D, H, L, P). Dot colors represent a model used for parameter estimation.

We found that the anisotropy in the spring constant of junction ($\mu_1$ and $\varphi_1$) exhibited more complex behaviors. In the wing, $\varphi_1$ was aligned along the AP axis, which is perpendicular to the axis of tissue stretching (Fig 6G and 6J). Model B, which omits $\mu_1$, was selected in 11 out of 23 samples at 16.5–18.5 h APF, when active cell rearrangement is dominant, whereas model A

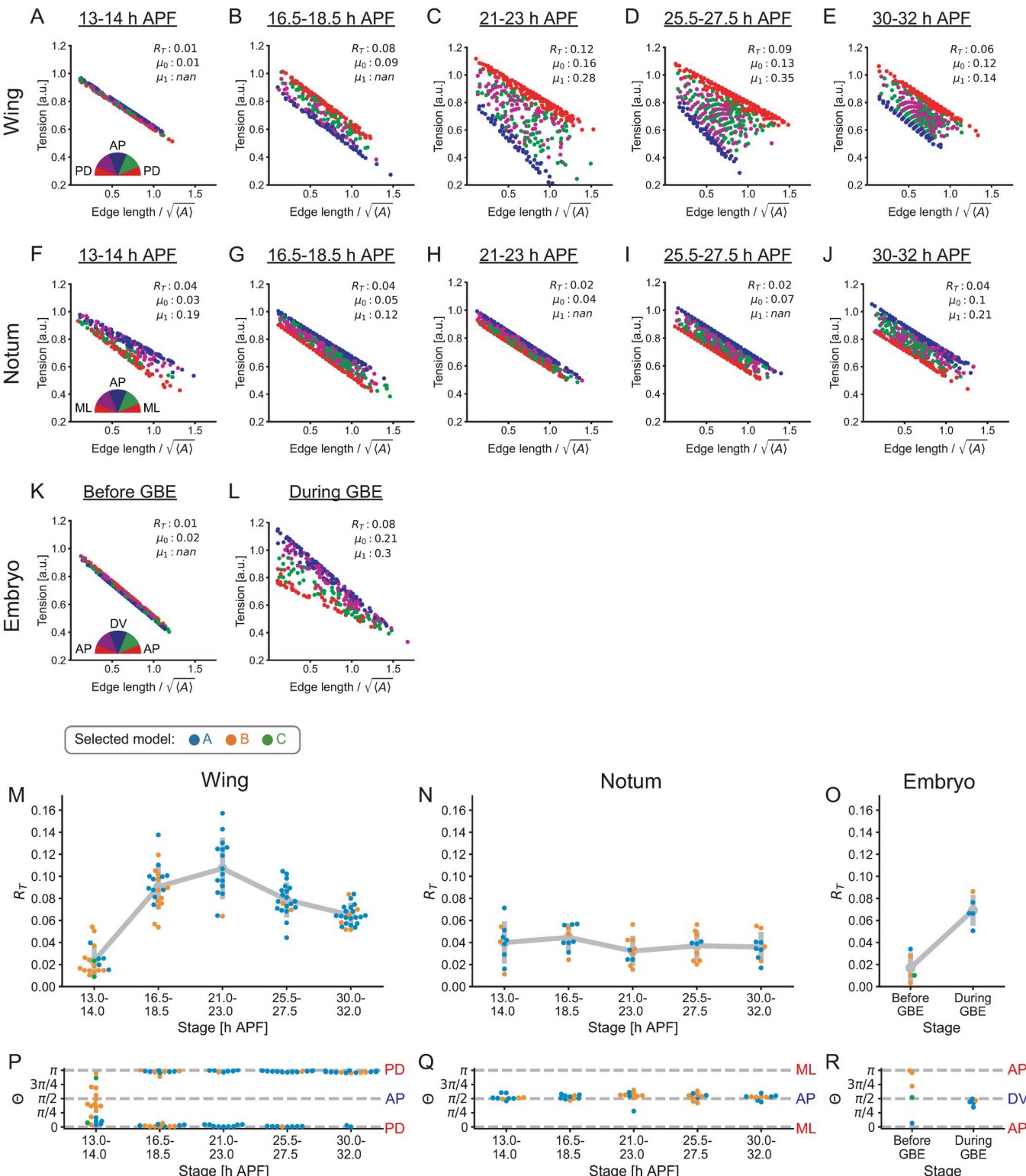

**Fig 5. Developmental changes in predicted tensions in *Drosophila* epithelial tissues.** (A–L) Junction tension predicted by estimated parameters is plotted against the junction length at the stage indicated in the pupal wing (A–E), pupal notum (F–J), and embryo (K, L). Estimated values of $R_T$, $\mu_0$ and $\mu_1$ are shown in the upper right corner. A semicircle indicates the classification of junctions based on their orientation for each tissue (PD: proximal-distal, AP: anterior-posterior, ML: medio-lateral and DV: dorsoventral). (M–R) Developmental changes in the anisotropy of the predicted tension ($R_T$; Materials and Methods) and its orientation ($\Theta$). Each dot shows the value of $R_T$ (M–O) or $\Theta$ (P–R) in each sample at the stage indicated in the pupal wing (M, P), pupal notum (N, Q), and embryo (O, R). The gray line connects the average values of $R_T$ for each stage. Error bars indicate the s.d. Dot colors represent a model selected by AIC.

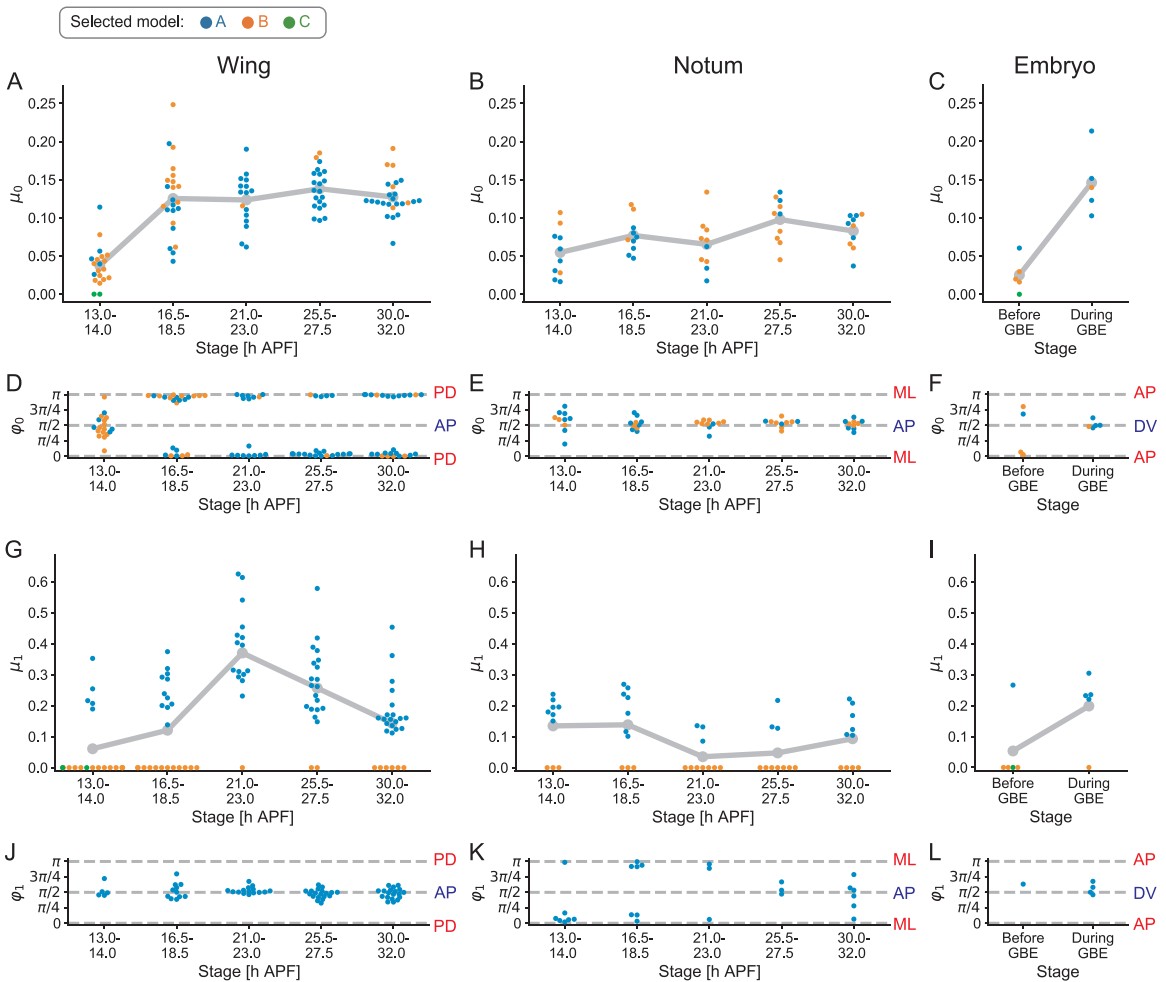

**Fig 6. Developmental changes in tension anisotropy parameters in *Drosophila* epithelial tissues.** (A–L) Developmental changes in the estimated values of $\mu_0$ (A–C; the anisotropy in the line tension), $\varphi_0$ (D–F; the orientation of anisotropy in the line tension), $\mu_1$ (G–I; the anisotropy in the spring constant of junction), and $\varphi_1$ (J–L; the orientation of anisotropy in the spring constant of junction). Each dot shows the estimated value of a parameter in each sample at the stage indicated in the pupal wing (A, D, G, J), pupal notum (B, E, H, K), and embryo (C, F, I, L). The gray line connects the average values of $\mu_0$ and $\mu_1$ for each stage. Dot colors represent a model selected by AIC.

was selected in 34 out of 37 samples associated with the increase in $\mu_1$ at 21–23 h APF and 25.5–27.5 h APF, when passive cell rearrangement proceeds (Fig 6G). In the notum, model A was selected in 26 out of 50 samples, and $\mu_1$ was smaller than that of the wing (Fig 6H and 6K). In the embryo, $\mu_1$ and $\varphi_1$ changed during development similarly to $\mu_0$ and $\varphi_0$ (Fig 6I and 6L). In summary, $\mu_1$, which represents the angular dependence of positive feedback between junction tension and shrinkage (see Model construction), increased during passive cell rearrangement in the wing and during active cell rearrangement in the embryo but not during active cell rearrangement in the wing. This could be explained by the fact that myo-II is concentrated at PD junctions to counteract extrinsic pulling from the hinge [46]; therefore, remodeling PD junctions does not strongly require upregulating tension/myo-II during active cell rearrangement in the wing (Figs 2D and 7A). On the other hand, passive cell rearrangement in the wing and active cell rearrangement in the embryo require a feedback mechanism to concentrate myo-II and elevate tension on remodeling junctions (Figs 2E and 2F and 7B and 7C and its legend).

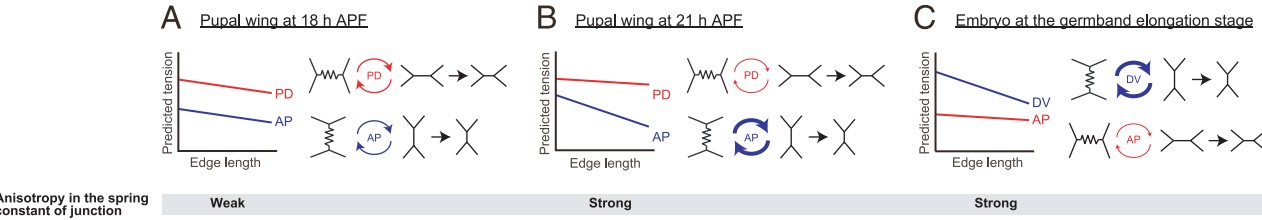

**Fig 7. Summary of the results of parameter estimation.** (A–C) Schematics showing the results of parameter estimation in the pupal wing at 18 h APF (A), in the pupal wing at 21 h APF (B), and in the embryo at the GBE stage (C). Line color represents junctions along the specific axis of the tissue and the arrow width represents the magnitude of the feedback. The results of parameter estimation indicated that the positive feedback between junction tension and shrinkage is strengthened at remodeling junctions during passive cell rearrangement in the pupal wing (B) and active cell rearrangement in the embryo (C) but not active cell rearrangement in the pupal wing (A).

## Tension anisotropy parameters are under the control of tissue patterning and stretching

To study the biochemical and mechanical mechanisms by which tension parameters are biased toward the specific axis of tissue, we examined how alterations in tissue polarity and stretching affect the anisotropy in tension parameters (Fig 8). First, we estimated mechanical parameters in the hinge-severed wing, where extrinsic pulling along the PD axis is removed, and directional bias in passive cell rearrangement is lost [45,46]. The tensions predicted from estimated parameter values in the hinge-severed wing exhibited a weaker directional bias in their basal value ($\mu_0$) and dependence on junction length ($\mu_1$) at 24 h APF (Fig 8A–8H; for the sake of comparison, we did not perform model selection and used model A throughout this section). As wing development proceeds, $R_T$, $\mu_0$, and $\mu_1$ decreased in the control wings (Fig 8C–8H). The difference in $\mu_1$ between the two groups diminished at 30–32 h APF when passive cell rearrangement almost ceased (Fig 8E). Second, we conducted parameter estimation on a loss-of-function mutant of a pair-rule gene *runt*, in which AP patterning of the embryo is disturbed and directional cell rearrangement is disrupted [53,57,58]. In the *runt* mutant, the anisotropy magnitude $R_T$, $\mu_0$, and $\mu_1$ significantly decreased, and the anisotropy orientations $\Theta$, $\varphi_0$, and $\varphi_1$ became more dispersed compared with the wild type (Fig 8I–8P). Collectively, our data suggest that tension anisotropy parameters are under the control of mechanical pulling in the wing and AP body patterning in the embryo and imply that decreases in tension anisotropy parameters could underlie cell rearrangement defects in the hinge-severed wing and in the AP patterning mutant embryo.

## The anisotropic spring model was selected over the conventional model

Finally, we examined whether the anisotropic spring model (model A in this study) or the conventional model [29,30] fits the experimental data better. We considered four nested conventional models (Model construction). The pressure function is the same in both classes of models, the conventional model and the anisotropic spring model. We performed model selection on the *Drosophila* pupal wing, pupal notum, and embryo and found that in all samples examined (n = 165), AIC selected the anisotropic spring model over the conventional model. For example, for the data shown in Fig 1E, AIC of model A was -261, whereas AIC of the conventional models ranged from 136 to 371; thereby model A was selected. The results suggest that the anisotropic spring model more appropriately reflects the mechanical nature of cell junctions in *Drosophila* epithelial tissues.

## Discussion

We developed a simple, two-stage statistical approach to estimate the mechanical parameters of epithelial tissue using only image data. In contrast to existing methods where parameter

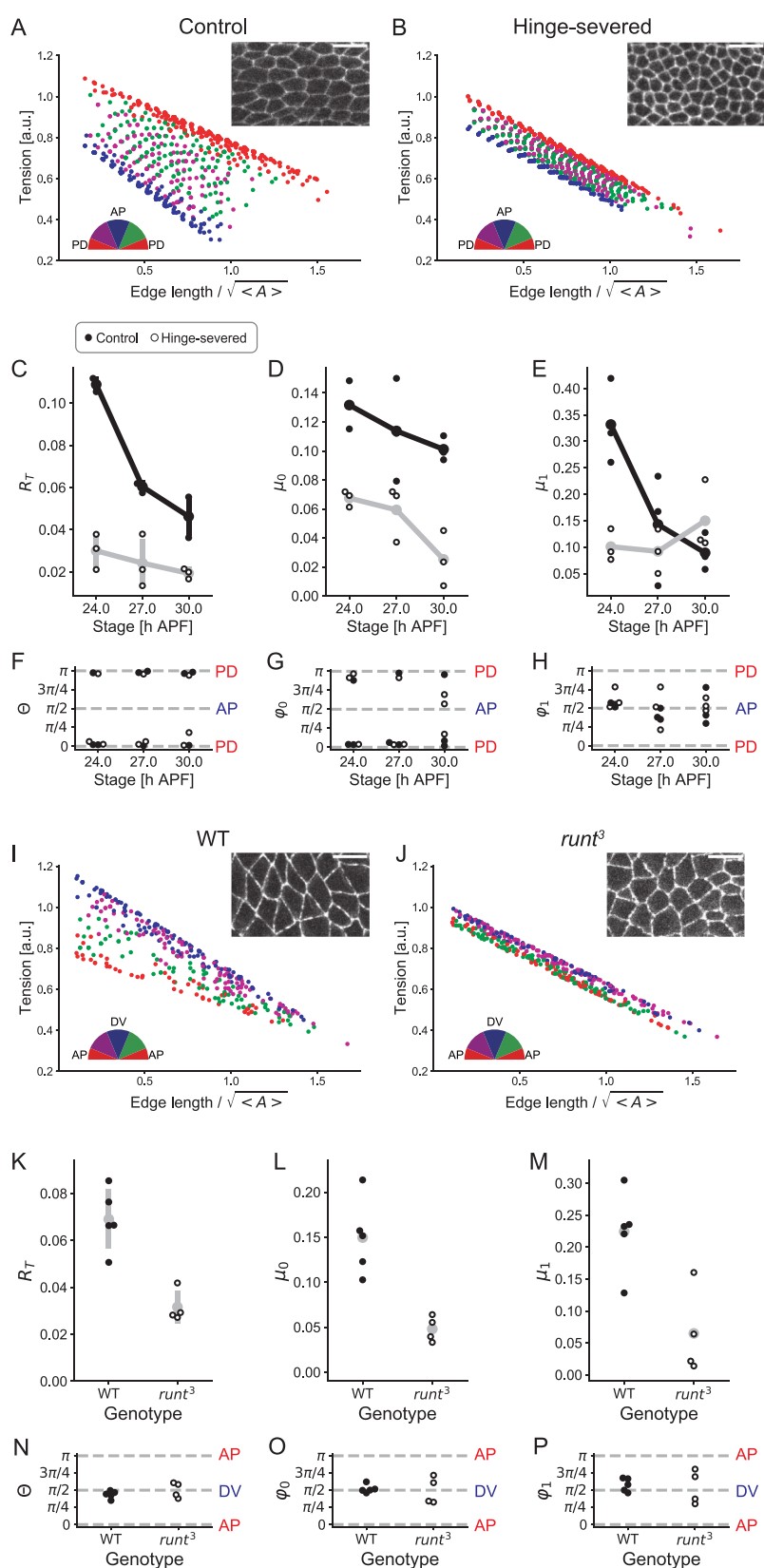

**Fig 8. Effect of alterations in tissue stretching of the wing and body polarity of the embryo on the anisotropy of tension parameters.** (A, B) Junction tension predicted by estimated parameters is plotted against the junction length for control and hinge-severed wings at 24 h APF. A semicircle indicates the classification of the junction based on its orientation. Insets in A and B are enlarged images of DE-cad-GFP in control and hinge-severed wings at 24 h APF. (C–H) $R_T$ (C; the anisotropy of the predicted tension), $\mu_0$ (D; the anisotropy in the line tension), $\mu_1$ (E; the anisotropy in the spring constant of junction), $\Theta$ (F; the orientation in anisotropy of the predicted tension), $\varphi_0$ (G; the orientation of anisotropy in the line tension), and $\varphi_1$ (H; the orientation of anisotropy in the spring constant of junction) from time-lapse images of control (black circles) and hinge-severed (open circles) wings. The black and gray lines connect the average estimated values of parameters for each stage in control and hinge-severed wings, respectively. (I, J) Junction tension predicted by estimated parameters is plotted against the junction length for wild-type and *runt*[3] embryos. Insets in I and J are enlarged images of DE-cad-GFP in wild-type and *runt*[3] embryos. (K–P) The anisotropy of the predicted tension ($R_T$; K), $\mu_0$ (L), $\mu_1$ (M), $\Theta$ (N), $\varphi_0$ (O), and $\varphi_1$ (P) in wild-type (black circles) and *runt*[3] (open circles) embryos at the GBE stage. Error bars indicate the s.d. Scale bars: 10 μm in (A, B) and 10 μm in (I, J).

fitting is done via summary statistics, our method directly estimates parameter values by comparing model equations with experimental data on cell shape and geometry. Our method does not require parameter sweeping by numerical simulations, and the estimation process is in most cases reduced to a linear regression problem. In addition, the low computational cost of the method allows us to statistically evaluate complex models, including models with multiple angular parameters. Thanks to this advantage of the method, we uncovered distinctive dynamics of the anisotropy of line tension and the anisotropy of the spring constant of the junction during morphogenesis, indicating the effectiveness of our approach.

Our formalism was tested for accuracy both *in silico* and *in vivo*. The average estimation error is less than several percent in the artificially generated data set, indicating that the parameter estimation method achieves a high level of precision *in silico*. In all tissues examined, the predicted tension is consistent with the known anisotropy of the tension. Moreover, estimated tension parameters can be interpreted along with experimental observations of cell behaviors and molecular localization. These results suggest that our method provides a reasonable estimate of tension parameters *in vivo*. On the other hand, the validity of pressure parameters remains unclear because knowledge about cell pressure in animal tissues is quite limited. Moreover, we observed large variance among cells in the pressure-cell area plot, which could lead to uncertainty in model construction. The *in vivo* evaluation of pressure functions/parameters awaits future developments in mechanical measurement techniques. Since the results in this study were not significantly affected by the choice of pressure function, we focus on tension parameters in the following discussion.

Thanks to the large number of data points in the tension-junction length plot obtained by using Bayesian force/stress inference, the angular dependency in the magnitude of the negative correlation between junction tension and length was detected. We then constructed tension functions on the basis of the quantitative data. Our analysis of the mechanical parameters of the model functions in *Drosophila* epithelial tissues suggests that the spring constant of the junction, which represents the magnitude of the feedback between junction tension and shrinkage, reflects the requirement to elevate tension/myo-II at remodeling junctions, which could facilitate junctional shrinkage. Furthermore, our data showing that the anisotropy in line tension and the anisotropy in the feedback are aligned along and perpendicular to the axis of tissue stretching during the late developmental stage of the wing, respectively, suggest that resistance to extrinsic pulling and junction remodeling are controlled via different tension parameters (Fig 7B). The molecular basis of the spatiotemporal dynamics of tension parameters is a target of future research.

It is important to recognize the limitations of the proposed method. First, it can provide only relative values of parameters because the parameter values are calculated from the force-balance equations at the cell vertex, which can be solved up to an unknown scale factor and

basal pressure. Additional experiments, such as optical tweezer measurements, are required to calibrate the estimated parameter values. Second, the parameter estimation method alone cannot distinguish underlying biophysical processes such as active vs. passive and intrinsic vs. extrinsic processes. Perturbation experiments are required to decompose estimated parameters into the effects of different processes. Third, there are limitations in model functions to be considered because multicollinearity, which is caused by the existence of strongly mutually correlated terms, may result in unstable parameter estimation [59]. This problem can be avoided by setting a prior function that limits the parameter space.

The current method can be extended to treat input data in different ways and/or incorporate different types of information. One may be interested in grouping samples by tissue, developmental stage, and/or genotype. For this purpose, hierarchical Bayesian formalism can be employed to infer a common meta-parameter set for each group and perform model selection based on the grouped data [60]. Hierarchical Bayesian formalism can also give mechanical parameters as a function of space and/or time [60]. To statistically treat the observation error of the vertex positions, one can formulate state-space modeling, by which the observation error is integrated into the estimation process. Future extensions of the parameter estimation method are expected to improve its accuracy and applicability.

Given that mechanics and genetics provide extensive mutual feedback to shape the tissue, combining mechanical measurement with a genetic approach is essential to advance our understanding of morphogenesis [1,61]. We believe that the simplicity of the proposed method will help us pursue this endeavor. For instance, it opens up the possibility of image-based, large-scale genetic screening for mutations that selectively affect specific mechanical parameters. The fact that our method could detect changes in the anisotropic tension parameters in genetically or surgically manipulated tissues suggests that it is a promising direction. We anticipate that the proposed method will provide a powerful tool to elucidate the comprehensive picture of morphogenesis.

## Materials and methods

### Bayesian force inference

Two of the authors previously developed Bayesian force/stress inference [36]. Here, we briefly explain its basic principle and assumptions. The details of the formalism and the results of *in silico* and *in vivo* validation can be found in [36,46,51,62].

Force/stress inference considers quasi-static force balance at the cell vertex along the 2D plane in epithelial tissue (Fig 1A–1C and Eq 4) and solves an inverse problem between cell shape and forces to obtain relative values of junction tension and cell pressure (Fig 1E and 1G) [36,63,64]. However, the problem is ill conditioned, *i.e.*, the number of conditions (twice the number of vertices) is less than the number of unknowns (the number of cell junctions plus the number of cells) when the curvature of the cell contact surface is small, but the cell pressure difference is not negligible in a tissue of interest. Bayesian force/stress inference treats the indefiniteness by incorporating our prior knowledge about the system. We tested different prior distributions and found that the one that assumes the positivity of junction tension yields accurate and robust estimates of forces [36,46,51,62].

Note that Bayesian force/stress inference and the parameter estimation method proposed in this study differ in the unknown number of equations to be solved. Bayesian force/stress inference solves the force-balance equations at the cell vertex without hypothesizing any mechanical model. It employs an empirical Bayes formulation to treat the indefiniteness and determines the most plausible solution [36]. On the other hand, the parameter estimation method solves the overdetermined force-balance equations with respect to the parameters of a hypothesized model to investigate the mechanical properties of cells.

## Parameter estimation

**Preprocessing.** We excluded outliers and normalized the spatial scale for the preprocessing of data. As the angle of a short junction is difficult to measure precisely, we excluded junctions shorter than 3 pixels (*in vivo* data) or $0.05 \times \sqrt{A_0}$ (synthetic data; $A_0$ is a target area). We also excluded cells larger than twice the median area of cells that appear in an input image, since significantly larger cells, including sensory cells in the notum, could possess different mechanical properties from other cells. After that, we normalized the spatial scale by the median cell area.

**Solution of force-balance equation.** For the parameter estimation of the anisotropic spring models, Eq 4 was solved by linear multiple regression as follows. The anisotropic coefficients $\Lambda_0(\theta_{ij})$ and $\Lambda_1(\theta_{ij})$ in Eq 2 can be expressed in linear form,

$$\Lambda_0(\theta_{ij}) = \lambda_0 + \lambda_0' sin2\theta_{ij} + \lambda_0'' cos2\theta_{ij},$$

$$\Lambda_1(\theta_{ij}) = \lambda_1 + \lambda_1' sin2\theta_{ij} + \lambda_1'' cos2\theta_{ij},$$

where $\mu_0$, $\varphi_0$, $\mu_1$, $\varphi_1$ in Eq 2 are transformed into
$\lambda_0' = \lambda_0\mu_0 sin2\varphi_0$, $\lambda_0'' = \lambda_0\mu_0 cos2\varphi_0$, $\lambda_1' = \lambda_1\mu_1 sin2\varphi_1$, $\lambda_1'' = \lambda_1\mu_1 cos2\varphi_1$. $T$ and $P$ are then written as $T = L_T\boldsymbol{\beta_T}$ and $P = L_P\boldsymbol{\beta_P}$, respectively, where matrices $L_T$ and $L_P$ are given as

$$L_T = \begin{pmatrix} \vdots & \vdots & \vdots & \vdots & \vdots & \vdots \\ 1 & sin2\theta_{ij} & cos2\theta_{ij} & -l_{ij} & -l_{ij}sin2\theta_{ij} & -l_{ij}cos2\theta_{ij} \\ \vdots & \vdots & \vdots & \vdots & \vdots & \vdots \end{pmatrix}, \quad L_P = \begin{pmatrix} \vdots \\ -A_i \\ \vdots \end{pmatrix}.$$

$\boldsymbol{\beta_T} = (\lambda_0, \ \lambda_0', \ \lambda_0'', \ \lambda_1, \ \lambda_1', \ \lambda_1'')^T$ and $\boldsymbol{\beta_P} = (k)$ are vectors of model parameters to be estimated. For the parameter estimation using model B–E, omitted parameters and corresponding columns were eliminated from $\boldsymbol{\beta_T}$ and $L_T$, respectively. Taken together, Eq 4 leads to an equation

$$F = CL\boldsymbol{\beta} = 0,$$

where we used

$$L = \begin{pmatrix} L_T & 0 \\ 0 & L_P \end{pmatrix}, \ \boldsymbol{\beta} = \begin{pmatrix} \boldsymbol{\beta_T} \\ \boldsymbol{\beta_P} \end{pmatrix}.$$

We estimated the parameters $\boldsymbol{\beta}$ by solving the equation. In practice, $\lambda_0$ is set to unity to fix a scaling factor of the solution, and other parameters are determined by minimizing $||F||^2$. Note that this is equivalent to a multiple linear regression problem, and the solution is unique up to the scaling factor determined by $\lambda_0$. We fixed $\lambda_0 = 1.0$. We set $0 < \mu_1 < 1.0$, because Bayesian force/stress inference analysis showed that the sign of the correlation between inferred junction tension and the junction length was not inverted with respect to the junction angle in all data analyzed in this study (*cf.*, Fig 5A–5L). We solved this problem and obtained AIC using the function **OLS.fit()** in the Python package statsmodels. After the estimation, we converted $(\lambda_0', \ \lambda_0'', \ \lambda_1', \ \lambda_1'')$ to $(\mu_0, \varphi_0, \mu_1, \varphi_1)$.

For the conventional model, the parameter $\Gamma$ is restricted to be positive [31,65]. Parameter estimation was thus performed by minimization of $||F||^2$ under the constraint $\Gamma > 0$ with the fixed scaling factor $\lambda_0 = 1.0$. We used **optimize.least_squares** in the Python package SciPy. After the estimation, AIC was calculated as Eq 5.

## The anisotropy of the predicted tension

The anisotropy of the predicted tension $R_T$ was calculated as

$$R_T \langle \hat{T}_{ij} \rangle e^{i2\Theta} = \langle \hat{T}_{ij} e^{i2\theta_{ij}} \rangle - \langle \hat{T}_{ij} \rangle \langle e^{i2\theta_{ij}} \rangle,$$

where $\hat{T}_{ij}$ is the predicted tension of the cell junction between the $i$-th and $j$-th cells and $\langle q_{ij} \rangle$ denotes the average value of $q_{ij}$ for the junctions analyzed [46]. $\Theta$ represents the orientation of $R_T$. $R_T$ decreases if the tension is uncorrelated with respect to the orientation of the junctions.

## *Drosophila* strains

The flies used in this study were *sqhp-sqh-GFP* (*sqh-GFP* expressed under the control of the natural *sqh* promoter [66]), *UAS-Dα-cat-TagRFP* [36], *apterous (ap)-Gal4*, *DE-cad-GFP* [67], and *runt³* (also known as *runt$^{LB5}$*) [53,68]. The fly genotypes were *DE-cad-GFP* (Figs 1E–1H and 1N–1Q and 4D and 4H and 4L and 4P and 5K and 5L and 5O and 5R and 6C and 6F and 6I and 6L and 8A–8I and 8K–8P and S1 and S4C and S4F), *sqhp-sqh-GFP*, *UAS-Dα-cat-TagRFP*/*sqhp-sqh-GFP*, *ap-Gal4* (Figs 4B and 4C and 4F and 4G and 4J and 4K and 4N and 4O and 5A–5J and 5M and 5N and 5P and 5Q and 6A and 6B and 6D and 6E and 6G and 6H and 6J and 6K and S4A and S4B and S4D and S4E), and *runt³/Y*; *DE-cad-GFP* (Fig 8J–8P).

## Surgical manipulation to relax tissue stretching

Pupal wings were detached from the hinge using forceps at 23 h APF as previously described [46,69].

## Image collection

Still images of the control wing and notum and time-lapse images of the control wing have been reported in [36,46] and [69], respectively. Other image data were acquired in this study.

*Drosophila* pupal wing and notum samples were prepared as previously reported [46,50,69,70]. Briefly, pupae were fixed to double-sided tape, and the pupal case above the left wing or the notum was removed. To conduct imaging of the wing, the pupae were mounted on a small drop of water or Immersol W 2010 (Zeiss 444969-0000-000) in a glass-bottom dish [46,50,69]. For imaging of the notum, the pupae were mounted on a glass slide. The chamber was sealed with silicon (Shin-Etsu Silicone, HIVAC-G) and a cover glass with a small drop of water [70]. Still images of the pupal wing and notum were acquired using an inverted confocal microscope (Olympus FV1000D) equipped with an Olympus 60x/NA1.2 SPlanApo water-immersion objective. Time-lapse images of the pupal wing were acquired at 1 min intervals from 24 h to 30 h APF using an inverted confocal spinning disk microscope (Olympus IX83 combined with Yokogawa CSU-W1) equipped with an iXon3 888 EMCCD camera (Andor), an Olympus 60×/NA1.2 SPlanApo water-immersion objective, and a temperature control chamber (TOKAI HIT) using IQ 2.9.1 (Andor) [50].

*Drosophila* embryos were prepared for live imaging as previously reported with some modifications [71]. Briefly, embryos were dechorionated using bleach and washed with water. Then, dechorionated embryos were mounted on a glass-bottom dish coated with glue. After making a bank surrounding the embryos from silicone grease (Shin-Etsu Silicone, HIVAC-G), the embryos were covered with silicone oil (Shin-Etsu Silicone, FL-100-1000CS). The outside of the bank was filled with water to prevent desiccation. Time-lapse images were acquired at 10 sec intervals using the abovementioned CSU imaging system equipped with an Olympus 60×/NA1.35 SPlanApo oil-immersion objective.

### Image processing and analysis

Image segmentation of the pupal wing and notum was performed using custom-made macro- and plug-ins in ImageJ as previously described [36]. EPySeg [72], a Python package for segmenting 2D epithelial tissues, was also used to segment cells from time-lapse images of the pupal wing and embryo. Vertex position and connectivity were extracted from skeletonized images by using custom-made code in OpenCV [36].

### Noise resistance test on the parameter estimation

To determine the robustness of the proposed method to image processing errors, noise to image data was included, and its effect on parameter estimation was quantified. Noised data were generated by adding Gaussian noise to 10% of the vertex coordinates in the original segmented data. The standard deviation of the noise was adjusted so that the relative changes in the components of the matrix, $CL$, were comparable between different data sets ($0.02 \times \sqrt{median(A)}$ and $0.05 \times \sqrt{median(A)}$ for synthetic and *in vivo* data sets, respectively). Parameter estimation on the noised data was conducted by using Model A (synthetic data set) or the model selected in the original data (*in vivo* data set). We calculated the deviations of the estimated parameter, $\delta q$, and the correlation coefficient, $r$, between the predicted tension from the original and the noised data. The deviation of the estimated parameter is defined as $\delta q = (q_n - q_t)/q_t$ for synthetic data set, where $q_t$ and $q_n$ denote the true value of a parameter and the estimated value from the noised data, respectively. For *in vivo* data set, the estimated parameter from the original data was used instead of the true value for the calculation of the deviation. The above procedure was repeated 100 times for each sample, and the median value of the deviation of the estimated parameter was plotted (Fig 4).

## Supporting information

**S1 Fig. Choice of pressure function does not affect the predicted tension.** (A–C) Tension predicted by estimated parameters is plotted against junction length. Dot color indicates the orientation of each junction relative to the PD axis of the wing (semicircle). Pressure functions used for estimation were $P(A_i) = -kA_i+P_0$ (A), $P(A_i) = k/A_i+P_0$ (B), and $P(A_i) = kexp(-A_i)+P_0$ (C). The plot in (A) is the same as that shown in Fig 1N.
(TIF)

**S2 Fig. The uncertainty of parameter estimation in synthetic data.** (A–D) The uncertainty quantification for $\lambda_0$ (A; the line tension), $\mu_0$ (B; the anisotropy in the line tension), $\lambda_1$ (C; the spring constant of junction), and $\mu_1$ (D; the anisotropy in the spring constant of junction). Synthetic data were generated from 20 different parameter sets. The standard error of the parameter was calculated from the residue and normalized by considering the error propagation based on the relationship described in the legend of Fig 3. Dots indicate the estimated value of the parameter. The error bar is defined as $[\hat{q} - 1.96\hat{\sigma}_q, \ \hat{q} + 1.96\hat{\sigma}_q]$, where $\hat{q}$ and $\hat{\sigma}_q$ are the estimated values of the normalized parameter and its standard error, respectively. The data showed that the error bars were narrow and included the true values of $\mu_0$ and $\mu_1$ in all samples and those of $\lambda_0$ and $\lambda_1$ in 12 of the samples.
(TIF)

**S3 Fig. *In silico* validation of model selection using AIC.** (A) Table of AIC values in synthetic data set. Synthetic data were generated using different models (data generation models), and AIC values were calculated for each model (estimation models). In each row, AIC values take the minimum value when the data generation model is used for inference (highlighted in

yellow).
(TIF)

**S4 Fig. Developmental changes in isotropy parameters in *Drosophila* epithelial tissues.** (A–F) Estimated values of $\lambda_1$ (A–C; the spring constant of junction) and $k$ (D–F; the elastic modulus of cells) at the stage indicated in the pupal wing (A, D), pupal notum (B, E), and embryo (C, F). The gray line connects the average estimated values of parameters for each stage. Dot colors represent a model selected by AIC.
(TIF)

## Acknowledgments

The authors would like to thank Yang Hong, Roger Kares, the Bloomington Stock Center, and the Kyoto Stock Center for fly strains; Olivier Lordereau and François Graner for foam images; Takashi Arai, Miho Aruga, Kyoko Komano, and Risa Matsui for technical assistance; Philippe Marcq for comments on the manuscript; and the iCeMS Analysis Center for imaging equipment.

## Author Contributions

**Conceptualization:** Goshi Ogita, Shuji Ishihara, Kaoru Sugimura.

**Investigation:** Goshi Ogita, Takefumi Kondo, Keisuke Ikawa, Kaoru Sugimura.

**Methodology:** Goshi Ogita, Shuji Ishihara.

**Software:** Goshi Ogita, Shuji Ishihara.

**Supervision:** Tadashi Uemura, Shuji Ishihara, Kaoru Sugimura.

**Visualization:** Goshi Ogita.

**Writing – original draft:** Goshi Ogita, Shuji Ishihara, Kaoru Sugimura.

**Writing – review & editing:** Takefumi Kondo, Keisuke Ikawa, Tadashi Uemura.

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
