## [Decision Letter · Decision Letter 0]

14 Jan 2022

Dear Dr. Sugimura,

Thank you very much for submitting your manuscript "Image-based parameter inference for epithelial mechanics" for consideration at PLOS Computational Biology. As with all papers reviewed by the journal, your manuscript was reviewed by members of the editorial board and by several independent reviewers. The reviewers appreciated the attention to an important topic. Based on the reviews, we are likely to accept this manuscript for publication, providing that you modify the manuscript according to the review recommendations.

Sincerely,

David M. Umulis

Associate Editor

PLOS Computational Biology

Jason Haugh

Deputy Editor

PLOS Computational Biology

[LINK]

Reviewer's Responses to Questions

**Comments to the Authors:**

Reviewer #1: In their manuscript “Image-based parameter inference for epithelial mechanics,” the authors formulated an image-based statistical method for estimating mechanical parameters of epithelial cellular networks. Then, they investigated candidate mechanical models by substituting the model functions into the force-balance equations and estimating model parameters with least-squares optimization. They verified the approach with synthetic data and applied the method to Drosophila epithelial tissues. The method identifies patterns in tension anisotropy correlated with extrinsic forces applied to the pupal wing and gene patterning within the embryo. The main finding of the work is inference of a more complex relationship between tension and cell-cell boundary lengths and angles. The method provides robust estimates of mechanical properties that can potentially be used for other contexts of epithelial morphogenesis. Their model selection approach has the potential to inform our understanding of the biophysical regulation of tissue morphogenesis.

In general the paper is clear and concisely written. Occasionally, the dense use of parameters detracts from readability. This could be ameliorated by reminding the reader of the physical interpretation of the parameters in each separate section and figure caption.

Readers who are not Drosophila wing experts would benefit from having Fig. S1 incorporated into the main text

In the authors' summary, it is claimed that the proposed methods give “more accurate estimates” than previous methods, but a systematic test of this claim is not provided. This could be more explicitly shown.

It would be very helpful for the community and the impact of the paper if the authors would publish their code on Github. It would be very helpful for transparency and verification (and to increase the potential impact of the work). Sharing the method/ code as an application would greatly enhance the perceived value to the field.

Minor typos:

Line 161: Add indefinite article: “In [a] tissue”

Line 388-389: Clarify: “same in both [classes] of models [non-conventional and conventional”

Reviewer #2: # Overview

Ogita et al. present a simple approach to estimating the mechanical properties of cells and tissues from image data. This approach is applicable to both in vitro and in vivo systems.

The approach presented is essentially a two stage estimation approach:

- First, forces and cell shapes are related via a previously established ‘Bayesian force/stress inference’ approach.

- Second, candidate constitutive equations are substituted into the above and parameters estimated via simple regression. Nested models can also be compared via e.g. AIC.

# General comments

I think this is a generally sensible approach. It relies heavily on the first stage ‘Bayesian force/stress inference’ method developed by two of the authors. The second stage is, as the authors note, essentially just linear/nonlinear regression.

My only real concern is that it is somewhat pushing it to call this a ‘new statistical method’, given the new, additional stage is simply regression. However, again, this is still a perfectly reasonable approach. I’m glad the authors include a synthetic data study.

A few comments/suggestions

- It would be good to include some measurement error in the synthetic data study. How robust are the results to this?

- Relatedly, regression methods allow simple confidence/credible intervals to be formed (taking the first stage as given, perhaps). It would be good to evaluate the performance of these on the synthetic data. E.g. given some measurement/imaging error, how often do the intervals trap the true parameters?

- The organisation of the article could perhaps be improved by combining ‘Experimental systems’, ‘A mechanical parameter estimation method’ and ‘Materials and methods’. These all seem like ‘Methods’ to me, and I found the division a bit confusing. It would also help to e.g. present the synthetic experimental system directly followed by the ‘real’ experimental systems within the same section. E.g. have ‘Experimental systems’ as part of ‘Materials and Methods’ and with ‘Synthetic’, ‘Pupal wing’, Pupal not’ and ‘Embryo’ as subheadings.

- In the first sentence I would just say ‘Mechanics plays many roles’ instead of ‘pleiotropic roles’.

- Line 164. I would delete ‘uniquely’ as other methods might exist.

- Line 399. ‘We developed a new’ statistical method to’ -> We developed a simple, two-stage statistical approach to’

**Have the authors made all data and (if applicable) computational code underlying the findings in their manuscript fully available?**

Reviewer #1: **No: **Code not available.

Reviewer #2: Yes

PLOS authors have the option to publish the peer review history of their article (what does this mean?). If published, this will include your full peer review and any attached files.

Reviewer #1: No

Reviewer #2: No

Figure Files:

Data Requirements:

Reproducibility:

References:

---

## [Decision Letter · Decision Letter 1]

7 Apr 2022

Dear Dr. Sugimura,

Thank you very much for submitting your manuscript "Image-based parameter inference for epithelial mechanics" for consideration at PLOS Computational Biology. As with all papers reviewed by the journal, your manuscript was reviewed by members of the editorial board and by several independent reviewers. The reviewers appreciated the attention to an important topic. Based on the reviews, we are likely to accept this manuscript for publication, providing that you modify the manuscript according to the review recommendations.

Sincerely,

David M. Umulis

Associate Editor

PLOS Computational Biology

Jason Haugh

Deputy Editor

PLOS Computational Biology

[LINK]

Reviewer's Responses to Questions

**Comments to the Authors:**

Reviewer #1: The revised version largely addressed issues raised in the first review.

A couple of remaining curiosities in the manuscript could be addressed in the final version:

1. The manuscript appears to be methodologically sound but the linked Github only provides parameter estimation code, not force inference. It could be more clearly described to enable full verification.

2. While several genotypes such as sqhp-sqh-GFP were reported in the study, it was not clear that sqh was quantified. Several claims about accumulation were stated, but it was not clear that this was shown. Additionally, ref. 66 did not seem to use the nomenclature sqhp and this should be defined.

Biological significance of findings could be highlighted better. There is a lack of quantification of Myosin, which could be clarified.

3. A summary figure S5 is helpful in understanding the main biological inferences made from the study. Burying it in the SI will obscure the key findings. The meaning of the color of the arrows in the right panel could be clarified.

Reviewer #2: The authors have essentially addressed all my comments and the article is a useful contribution to the literature. However I would note that the comments like

> 95% confidence intervals included the true value in most cases (Supplementary Fig. 2).

are not quite precise. In particular, if an interval procedure does not generate a close to 95% coverage rate in a synthetic experiment with sufficiently many repetitions then it can't really be called a 95% confidence interval in the frequentist sense -- 95% coverage being the defining property of a 95% confidence interval. However, it is well known that many approximate confidence interval procedures don't exactly satisfy their intended (or nominal) coverage. This is a long way of saying that it would be better to say e.g.

> The approximate 95% confidence intervals included the true value in most cases (Supplementary Fig. 2).

Finally, I'm wondering if any noise was added during the synthetic data experiment -- the intervals are very narrow. If there was and the noise only had a very small impact on the estimates that's fine, but surprising. If no noise was added then I'm wondering how 95% confidence intervals in the frequentist sense (i.e. based on a sampling distribution under a noise model) were formed. Perhaps these are more like Bayesian credible intervals?

Most of this is nit-picking based on supplementary material, but I think being a bit more precise in how these things are discussed would increase the chance that the method can be appropriately used by others on real-world problems.

**Have the authors made all data and (if applicable) computational code underlying the findings in their manuscript fully available?**

Reviewer #1: **No: **See comments above.

Reviewer #2: Yes

PLOS authors have the option to publish the peer review history of their article (what does this mean?). If published, this will include your full peer review and any attached files.

Reviewer #1: No

Reviewer #2: No

Figure Files:

Data Requirements:

Reproducibility:

References:

---

## [Editor Report · Decision Letter 2]

17 May 2022

Dear Dr. Sugimura,

We are pleased to inform you that your manuscript 'Image-based parameter inference for epithelial mechanics' has been provisionally accepted for publication in PLOS Computational Biology.

Best regards,

David M. Umulis

Associate Editor

PLOS Computational Biology

Jason Haugh

Deputy Editor

PLOS Computational Biology

---

## [Editor Report · Acceptance letter]

31 May 2022

PCOMPBIOL-D-21-01805R2 

Image-based parameter inference for epithelial mechanics

Dear Dr Sugimura,

I am pleased to inform you that your manuscript has been formally accepted for publication in PLOS Computational Biology. Your manuscript is now with our production department and you will be notified of the publication date in due course.

With kind regards,

Zsofia Freund
